# Identification of microRNAs involved in NOD-dependent induction of pro-inflammatory genes in pulmonary endothelial cells

Ann-Kathrin Vlacil[1], Evelyn Vollmeister[2], Wilhelm Bertrams[2], Florian Schoesser[1], Raghav Oberoi[1], Jutta Schuett[1], Harald Schuett[1], Sonja Huehn[3], Katrin Bedenbender[2], Bernd T. Schmeck[2,4,5,6], Bernhard Schieffer[1], Karsten Grote[1]*

1 Cardiology and Angiology, Philipps-University Marburg, Marburg, Germany, 2 Institute for Lung Research/ iLung, German Center for Lung Research, Universities of Giessen and Marburg Lung Center, Philipps-University Marburg, Marburg, Germany, 3 Department of Hematology, Oncology, and Immunology, Philipps-University Marburg, Marburg, Germany, 4 Department of Medicine, Pulmonary and Critical Care Medicine, University Medical Center Marburg, Philipps-University Marburg, Marburg, Germany, 5 Center for Synthetic Microbiology (SYNMIKRO), Philipps-University of Marburg, Marburg, Germany, 6 German Center for Infection Research (DZIF), partner site Giessen-Marburg-Langen, Marburg, Germany

* karsten.grote@staff.uni-marburg.de

**Data Availability Statement:** All relevant data are within the manuscript and its Supporting Information files.

## Abstract

The nucleotide-binding oligomerization domain-containing proteins (NOD) 1 and 2 are mammalian cytosolic pattern recognition receptors sensing bacterial peptidoglycan fragments in order to initiate cytokine expression and pathogen host defense. Since endothelial cells are relevant cells for pathogen recognition at the blood/tissue interface, we here analyzed the role of NOD1- and NOD2-dependently expressed microRNAs (miRNAs, miR) for cytokine regulation in murine pulmonary endothelial cells. The induction of inflammatory cytokines in response to NOD1 and NOD2 was confirmed by increased expression of tumour necrosis factor (*Tnf*)-*α* and interleukin (*Il*)-*6*. MiRNA expression profiling revealed NOD1- and NOD2-dependently regulated miRNA candidates, of which miR-147-3p, miR-200a-3p, and miR-298-5p were subsequently validated in pulmonary endothelial cells isolated from *Nod1/2*-deficient mice. Analysis of the two down-regulated candidates miR-147-3p and miR-298-5p revealed predicted binding sites in the 3' untranslated region (UTR) of the murine *Tnf-α* and *Il-6* mRNA. Consequently, transfection of endothelial cells with miRNA mimics decreased *Tnf-α* and *Il-6* mRNA levels. Finally, a novel direct interaction of miR-298-5p with the 3' UTR of the *Il-6* mRNA was uncovered by luciferase reporter assays. We here identified a mechanism of miRNA-down-regulation by NOD stimulation thereby enabling the induction of inflammatory gene expression in endothelial cells.

## Introduction

The nucleotide-binding oligomerization domain-containing (NOD) proteins NOD1 (previously known as caspase recruitment domain family member 4, CARD4) and NOD2 (CARD15) are the central members of the intracellular NOD-like receptor family belonging to the

**Funding:** B.T.S. (031L0140) German Federal Ministry of Education and Research (BMBF, ERACoSysMed) - https://www.bmbf.de/en/index.html B.S. and K.G. (62-0002) Behring Röntgen foundation - https://www.br-stiftung.de/ The funders did not play any role in the study design, data collection and analysis, decision to publish, or preparation of the manuscript.

**Competing interests:** The authors have declared that no competing interests exist.

superfamily of pattern-recognition receptors (PRRs) of the innate immune system. NOD1 and NOD2 have a similar structure consisting of three functionally different domains: 1) a C-terminal leucine-rich-repeat (LRR) domain for ligand binding, 2) a central nucleotide-binding oligomerization domain (NBD) and 3) a N-terminal caspase activation and recruitment domain (CARD) for signalling. As cytosolic sensors, they lack a transmembrane domain and are responsible for the recognition of conserved motifs in bacterial peptidoglycan (so-called muropeptides) of both Gram-positive and Gram-negative bacteria and the subsequent initiation of inflammatory and anti-microbial responses [1,2]. NOD1 and NOD2 are not only expressed in immune cells but also in tissue cells including vascular endothelial cells [3,4]. Endothelial cells line the inner lumen of blood and lymphatic vessels but also form an immense capillary network. Due to their exposed location, this cell type is of particular importance for pathogen recognition at the interface between blood and diverse tissues and organs in the body. During sepsis or other infectious scenarios, the endothelium is massively attacked by pathogenic bacteria and their components are actively involved in the subsequent immune response by expressing and releasing inflammatory cytokines, controlling coagulation as well as leucocyte attraction and trafficking [5]. In recent years, different routes of entry and processing of bacterial peptidoglycan into the cytoplasm of host cells to activate NOD-dependent signalling have been explored [1].

In this regard, NOD1 and NOD2 show many similarities. In both cases, ligand binding activates the CARD-domain-containing serine/threonine kinase receptor-interacting protein 2 (RIP2, also known as RICK) [6] and subsequently nuclear factor-kappa B (NF-κB) and mitogen-activated protein kinase (MAPK) [7,8]. This in turn is responsible for an adequate expression of inflammatory cytokines in order to orchestrate the cellular immune response [9]. Beside transcription factors and epigenetic mechanisms microRNAs (miRNAs, miR) represent an additional diverse and largely overlapping system in the regulation of gene expression. MiRNAs are non-coding RNA species of 20–24 nucleotides, which target specific mRNAs via binding to their 3' untranslated region (UTR) thereby controlling posttranscriptional protein expression [10].

Only few studies have addressed a link between miRNAs and NOD so far and are focused mostly on the regulation of NOD expression itself by certain miRNAs. For example, NOD1 is targeted by miR-146a in γδ T cells [11] and by miR-495 in cardiac fibroblasts [12], whereas NOD2 is e.g. targeted by miR-10a in inflamed mucosa tissue and dendritic cells [13] and by miR-320 in colon cancer cells [14] with effects on cytokine expression down-stream of NOD. However, so far only one recent study investigated a link between miRNAs and NOD1 in endothelial cells. In this study, Kang et al. reported that miR-125a directly targets NOD1 and thereby controls the NOD1-dependent induction of angiogenesis [15]. The induction of inflammatory genes such as interleukin (Il)-1β, Il-6, Il-8 and type I interferons in endothelial cells has already been established in response to NOD1 and NOD2 ligands [16–18], suggesting that the endothelial NOD-pathway plays a role in the anti-infection immune response. We here isolated primary murine pulmonary endothelial cells to identify miRNAs responsible for the regulation of inflammatory genes in response to NOD1 and NOD2 stimulation.

## Material and methods

### Reagents and antibodies

l-Ala-γ-D-Glu-meso-diaminopimelic acid (TriDAP), muramyl dipeptide (MDP) and polyinosinic-polycytidylic acid (polyI:C) were purchased from Invivogen (San Diego, CA, USA). Macrophage-activating lipopeptide of 2 kDa (MALP-2) was synthesized and purified as described before [19]. Lipopolysaccharide (LPS) and isolectin B4 were from Sigma-Aldrich (Seelze,

Germany), 4',6-Diamidine-2'-phenylindole dihydrochloride (DAPI) from Life Technologies™ (Darmstadt, Germany), 4% formalin (Roti®-Histofix) from Carl Roth (Karlsruhe, Germany). Rat anti-mouse CD31 and CD144 antibodies were obtained from BD Biosciences (San Jose, CA), Dynabeads sheep anti-rat IgG was purchased from Invitrogen (Carlsbad, CA, USA). VectaMount permanent mounting medium was used from Vector Laboratories (Burlingame, CA, USA).

## Cells

Primary endothelial cells were isolated from lungs and hearts of male C57BL/6N wild type mice or $Nod1^{-/-}$, $Nod2^{-/-}$ double knockout (Nod1/Nod2-KO) mice on a C57BL/6N background (#50/2015). All experiments were approved by the governmental animal ethics committee and are conform to the guidelines from directive 2010/63/EU of the European Parliament. For every isolation, three mice at the age of six to ten weeks were sacrificed by cervical dislocation. The skin was soaked in 70% ethanol and lobes of the lung were removed under a sterile laminar flow hood and placed in 15 mL cold DMEM (Dulbecco´s Modified Eagle Medium)/F-12, (Life Technologies™), 20% fetal calf serum (FCS, PAN-Biotech, Aidenbach, Germany), 1% penicillin/streptomycin (100 U/mL and 100 mg/mL, Sigma-Aldrich) on ice. After dissection, lung lobes were collected in a 6 cm dish and subsequently minced with scissors for 5 minutes. Afterwards, minced lung tissue was transferred to 25 mL pre-warmed collagenase (2 mg/mL, Worthington Biochemical Corporation, Lakewood, NJ, USA) in Hanks' balanced salt solution (Sigma-Aldrich) and incubated for 45 minutes at 37°C with gentle agitation (100 rpm). Using a 12G cannula, the suspension was triturated 12 times and afterwards pipetted through a 70 μm disposable cell strainer (Greiner Bio-One, Germany). Further digestion was stopped by adding DMEM/F-12, 20% FCS, 1% P/S and cells were centrifuged for 10 minutes at 4°C and 300 g. The pellet was resuspended in 2 mL phosphate-buffered saline (PBS), 0.1% bovine serum albumin (BSA, Sigma-Aldrich).

Purified rat anti-mouse CD144 antibody was previously incubated overnight with dynabeads sheep anti-rat IgG in PBS, 0.1% BSA. For each isolation, 30 μL beads ($4x10^8$ beads/mL) and 6 μL antibody (0.5 mg/mL) were used. Cells were incubated with CD144 bound to magnetic beads for 15 minutes at RT on a rotator at 10 rpm. Afterwards, cells were washed in a magnetic separator (Merck Millipore, Burlington, MA, USA) with DMEM/F-12, 20% FCS, 1% P/S until the supernatant was clear. Cells were resuspended and incubated in DMEM/F-12, 20% FCS, endothelial cell growth supplement/heparin (12 μg/mL, ECGS/H, Promocell, Heidelberg, Germany), 1% P/S in fibronectin (1 μg/mL, Promocell) coated T75 flasks (Sarstedt, Nümbrecht, Germany) at 37°C and 5% $CO_2$ until cells reached confluence. The day after isolation, cells were washed three times with warm DMEM/F-12, 20% FCS, 1% P/S to remove loosely adherent cells. After reaching confluence, cells were detached by trypsin-ethylene diamine tetraacetic acid (trypsin-EDTA, 0.05%, Life Technologies™), washed, pelleted and incubated again in 2 mL PBS, 0.1% BSA with magnetic bead-bound purified rat anti-mouse CD31 antibody for 15 minutes at RT on a rotator at 10 rpm. After three times washing with DMEM F-12, 20% FCS, 1% P/S on a magnetic separator cells were cultured in fibronectin-coated 6-well plates (Eppendorf, Hamburg, Germany) in DMEM/F-12, 20% FCS, ECGS-H (12 μg/mL), 1% P/S and used for further experiments when reaching confluence. Medium was changed every second day.

The use of human umbilical cords from healthy volunteers was approved by the local ethic committee (AZ 20/16, Philipps University Marburg). Human umbilical vein endothelial cells were isolated and cultivated as described previously [20] with following modifications. Umbilical cords were flushed with Hank´s balanced salt solution with magnesium and calcium (HyClone™, GE Healthcare, Solingen, Germany) and PBS with magnesium chloride and

calcium chloride (Merck KGaA, Darmstadt, Germany). Endothelial cells were isolated by administration of 0.1% collagenase D solution from *Clostridium histolyticum* (Merck KGaA) into the vein for 20 minutes at 37°C and subsequently cultured in endothelial cell growth medium purchased from PromoCell supplemented with 1% penicillin/streptomycin and used up to passage 4 for all indicated experiments.

Human embryonic kidney (HEK)-293T cells, used for luciferase reporter assay, were purchased from ATCC (CRL-3216, LGC Standards GmbH, Wesel, Germany) and cultured in DMEM (Sigma-Aldrich) with 10% fetal bovine serum (Superior, Biochrom GmbH, Berlin, Germany).

## Stimulation of endothelial cells with NOD agonists

After reaching confluence, primary endothelial cells were starved for 2 hours in DMEM/F-12, 2% FCS, 1% penicillin/streptomycin and human umbilical vein endothelial cells for 2 hours in endothelial cell basal medium with 1% penicillin/streptomycin and stimulated with the NOD1 agonists Tri-DAP (10 μg/mL) and the NOD2 agonist MDP (10 μg/mL) for 6 and 24 hours. Subsequently, supernatants were collected and cells were lysed for RNA isolation in RNA--Solv® Reagent (Omega Bio-tek, Norcross, GA, USA).

## Transfection of endothelial cells and HEK-293T cells

Pulmonary endothelial cells were transfected with microRNAs mimics or anti-miRNA (miRI-DIAN microRNA hairpin inhibitor) using Lipofectamine® RNAiMAX Reagent (Invitrogen) according to the manufacturer's instructions. Briefly, mimics for murine miR-147-3p or miR-298-5p or anti-miR-298-5p and Lipofectamine® RNAiMAX Reagent were resuspended with Opti-MEM® Medium (Gibco, Waltham, MA, USA) and incubated for 5 minutes. MicroRNA/lipid complexes were added to the cells and incubated for 24 hours and final miRNA concentration used per well was 25 pmol. Transfection with microRNA mimic negative control served as control. MicroRNA mimics and anti-miRNAs were obtained from Dharmacon™ (Lafayette, CO, USA).

For Luciferase reporter assay, $1.3 \times 10^5$ HEK-293T cells per cm$^2$ were reverse transfected with Lipofectamine® 2000 (Thermo Fisher Scientific) following manufacturer's instructions using 200 ng psiCHECK™ 2 (Promega) as empty vector control or psiCHECK2 containing *Tnf-α* 3' UTR or *Il6* 3' UTR and 30 pmol miRNA scramble negative control, miRNA-147-3p or miR-298-5p mimics (Dharmacon) and incubated for 72 hours in DMEM/OptiMEM medium (Sigma-Aldrich/Thermo Fisher Scientific) prior lysis.

## Real-time PCR

For the analysis of mRNA expression, total RNA from endothelial cells was isolated using RNA-Solv® Reagent following the manufacturer's instructions and reverse-transcribed with SuperScript reverse transcriptase, oligo(dT) primers (Thermo Fisher Scientific, Waltham, MA), and deoxynucleoside triphosphates (Promega, Mannheim, Germany). Real-time PCR was performed in duplicate in a total volume of 20 μL using Power SYBR green PCR master mixture (Thermo Fisher Scientific) or real-time PCR probes and TaqMan Fast Advanced MasterMix (Thermo Fisher Scientific) on a Step One Plus real-time PCR system (Applied Biosystems, Foster City, CA) in 96-well PCR plates (Applied Biosystems). SYBR Green or FAM/BHQ-1 fluorescence emissions were monitored after each cycle. For normalization, expression of GAPDH was determined in duplicates. Relative gene expression was calculated by using the $2^{-\Delta\Delta Ct}$ method. Real-time PCR primers and probes were obtained from Microsynth AG

(Balgach, Switzerland) and are available upon request. Modification of real-time PCR probes: 5´ = FAM, 3´ = BHQ-1.

For the analysis of miRNA expression, total RNA was isolated using RNA-Solv® Reagent and reverse-transcribed with TaqMan MicroRNA Reverse Transcription Kit and TaqMan MicroRNA Assays for each miRNA (Applied Biosystems). Real-time PCR was performed in a total volume of 20 μL on a Step One Plus real-time PCR system (Applied Biosystems) in 96-well PCR plates (Applied Biosystems). FAM/MGB-NFQ fluorescence emissions were monitored after each cycle. For normalization, expression of U6 and sno202 was determined. Relative gene expression was calculated by using the $2^{-\Delta\Delta Ct}$ method. MicroRNA PCR primers and probes were obtained from Thermo Fisher Scientific and are available upon request. Modification of real-time PCR probes: 5´ = FAM, 3´ = MGB-NFQ.

## Enzyme-linked immunosorbent assay (ELISA)

Supernatant of confluent pulmonary endothelial cells were analyzed for TNF-$\alpha$ and IL-6 by using a mouse-specific ELISA from R&D Systems (Minneapolis, MN, USA) according to manufacturer's protocol with the help of an Infinite M200 PRO plate reader (TECAN Instruments, Maennedorf, Switzerland).

## TaqMan low-density array (TLDA)

Concentration and quality of total RNA isolations using a phenol-chloroform extraction with Isol-RNA lysis reagent (5´Prime, Hamburg, Germany) from endothelial cells were verified by a NanoDrop™ 1000 (Peqlab VWR, Radnor, PA) and an Agilent 2100 bioanalyzer (Agilent, Santa Clara, CA, USA) using the Agilent RNA 6000 Nano kit. In each case, 4 isolations of control, 6 hours TriDAP and 6 hours MDP with a 260nm/280nm ratio of at least 1.80 and RNA integrity numbers (RIN) of at least 9.0 were selected for subsequent TLDA analysis (Applied Biosystems).

For cDNA synthesis, 350 ng of total RNA were subjected to reverse transcription using Taqman MicroRNA Reverse Transcription Kit and the Megaplex RT Primers for rodent Pool A following manufacturer's instructions. RT was performed on a peqSTAR 2xGradient (PeqLab Biotechnologie GmbH, Erlangen, Germany) with following cycling conditions: 40 cycles of 16˚C for 2 min, 42˚C for 1 min and 50˚C for 1 sec, followed by 85˚C for 5 min and cooling to 4˚C. RT-PCR reaction was mixed according to manufacturer's protocol using TaqMan Fast Advanced Mastermix. Taqman Low Density Array (TLDA; Set A, v. 2.0, rodent, 384-well format, Life Technologies) was handled according to manufacturer's instructions with a centrifugation step 300xg for 2x1 min on a Heraeus™ Multifuge™ X3R (Thermo Fisher Scientific). TLDAs were executed on a ViiA 7 real-time PCR system (Applied Biosystems) with following cycling conditions: 50˚C for 2 min, 92˚C for 10 min, followed by 40 cycles of 97˚C for 1 sec and 62˚C for 20 sec. The cycle threshold was automatically set up by SDS 2.3 software (Thermo Fisher Scientific). For quantification of relative expression levels comparative $\Delta\Delta C_t$ method was used and fold changes were calculated by expression $2^{-\Delta\Delta Ct}$. Total array data are available at the National Center for Biotechnology Information's (NCBI) Gene Expression Omnibus (GEO, GSE145798).

## miRNA target site analysis

Predicted binding sites in the 3' UTR of the *Tnf-α* and *Il-6* mRNA sequence for miR-147-3p and miR-298-5p were identified with the freely accessible TargetScan and MirTarBase platform or by searching seed sequences in 3' UTR (https://www.ensembl.org; http://www.targetscan.org/vert_72/; http://mirtarbase.mbc.nctu.edu.tw/php/index.php).

## Plasmid generation

The sequences of full length 3' UTRs of murine wildtype *Tnf-α* (765 bp) and *Il-6* (429 bp) mRNAs were obtained from Ensembl (Release 92 April 2018). Mutations of putative miRNA binding sites were introduced as deletions of corresponding seed sequences. `5'-GTTT` and XhoI (`5'-CTCGAG`) as well as NotI (`GCGGCCGC-3'`) restriction sites and TTTG-3' overhang were added and purchased as a gBlocks gene fragment from Integrated DNA Technologies (Skokie, Il). Insertion into the psiCHECK™-2 vector was done via XhoI and NotI restriction sites and T4 DNA ligase. After transformation into One Shot™ TOP10 chemically competent *E. coli* (Thermo Fisher Scientific) correctness of the integration was verified via sequencing (sequencing service, Ludwig-Maximilians-University, Munich, Germany).

## Luciferase reporter assay

For detection of luciferase activity, Dual-Glo® Luciferase Assay System Kit (Promega) was used according to the manufacturer's instructions. Briefly, HEK-293T cells were transfected with the respective vector construct and miRNA and were lysed after 72 h. Substrates for Renilla and Firefly luciferase were added and luminescence was measured using a Tecan Infinite M200 Pro plate reader (TECAN Instruments).

## Statistical analysis

All data are represented as means + SEM. Data were compared using the 2-tailed Student t-test for independent samples or by 1-way ANOVA followed by Tukey multiple comparison test (GraphPad Prism, version 6.05; GraphPad Software, La Jolla, CA, USA). A value of $P<0.05$ was considered statistically significant. Numbers of independent experiments are indicated in each figure legend. Real-time PCR and ELISA was performed in technical duplicates.

Differential microRNA expression analysis from the TLDA dataset was performed with R (version 3.2.2) with the package limma (version 3.24.15) and the package HTqPCR (version 3.2) [21]. Expression was normalized to the expression of U6.

## Results

### *Nod1* and *Nod2* expression in murine pulmonary endothelial cells

Endothelial cells are highly relevant for anti-infection immune response since they are directly located at the interface of blood and the subjacent tissue. To identify NOD-dependently differentially expressed endothelial miRNAs, we initially isolated endothelial cells from lung tissue of wild type mice by CD144/CD31-based magnetic bead separation. Isolated cells were grown to confluent monolayers and were highly positive for the endothelial marker isolectin B4 in immunofluorescence (S1A Fig) and CD31 in flow cytometry (S1B Fig). Furthermore, they show substantial expression of *Nod1* and *Nod2* mRNA (S1C Fig) confirming these isolated pulmonary endothelial cells as a suitable target for NOD ligands and NOD-dependent signaling.

### Induction of pro-inflammatory genes in response to NOD1 and NOD2 stimulation in different endothelial cells

Next, we stimulated pulmonary endothelial cells with the NOD1-specific agonist Tri-DAP and the NOD2-specific agonist MDP (each 10 μg/mL) and exemplarily investigated gene expression of the major inflammatory NF-κB-dependent genes *Tnf-α* and *Il-6*. Real-time PCR analysis revealed induced mRNA expression of the NF-κB-dependent genes *Tnf-α* and *Il-6* after 6 hours, which returned to baseline levels after 24 hours (Fig 1A) confirming that NOD-

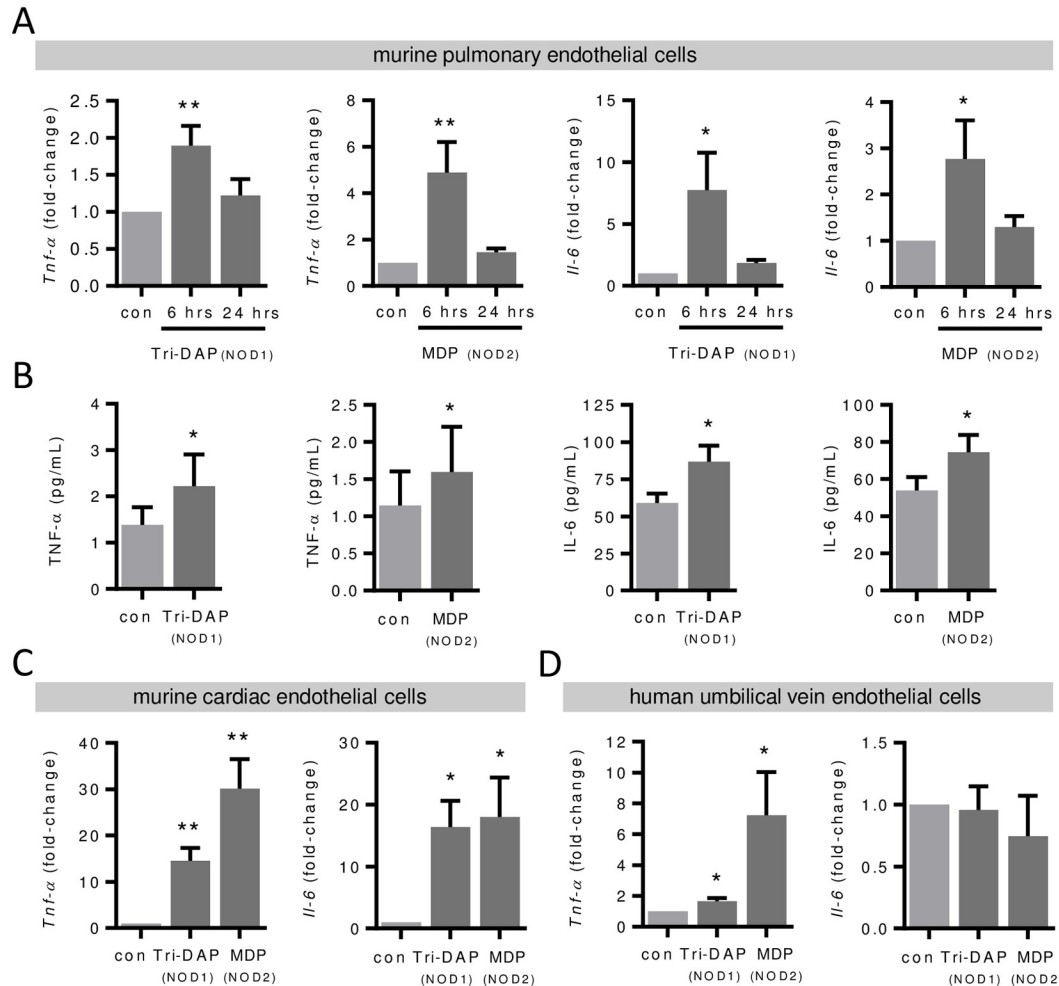

**Fig 1. Induction of pro-inflammatory genes in response to NOD1 and NOD2 stimulation in different endothelial cells.**
Murine pulmonary endothelial cells were stimulated with Tri-DAP or MDP (each 10 μg/mL) and (**A**) *Tnf-α* and *Il-6* mRNA levels were analyzed after 6 and 24 hours by real-time PCR and (**B**) TNF-α and IL-6 protein levels in the supernatant after 24 hours by ELISA. *P<0.05, **P<0.01 vs. unstimulated control (con), n = 5–8. *Tnf-α* and *Il-6* mRNA levels following Tri-DAP or MDP (each 10 μg/mL) stimulation in (**C**) murine cardiac endothelial cells and *Tnf-α* and *Il-6* mRNA levels in (**D**) human umbilical vein endothelial cells were analyzed after 6 hours by real-time PCR. *P<0.05, **P<0.01 vs. unstimulated control (con), n = 4.

signaling induced a pro-inflammatory gene expression in pulmonary endothelial cells. In addition, we analyzed cell culture supernatants after NOD stimulation for secreted proteins by ELISA. In accordance with the PCR analysis, we detected increased TNF-α and IL-6 protein levels after NOD stimulation (Fig 1B). To investigate whether NOD-dependent induction of pro-inflammatory expression is a general mechanism in endothelial cells, we additionally tested endothelial cells of other origin and species. We detected increased *Tnf-α* and *Il-6* mRNA levels after 6 hours of NOD stimulation in murine cardiac endothelial cells (Fig 1C). NOD stimulation increased *TNF-α* but not *IL-6* mRNA levels in human umbilical vein endothelial cells (Fig 1D). Both endothelial cell types showed *Nod1* and *Nod2* mRNA expression (S1D and S1E Fig).

## NOD1- and NOD2-dependently regulated miRNAs in murine pulmonary endothelial cells

The main purpose of this study was to identify miRNA candidates, which are involved in the regulation of this pro-inflammatory gene expression. To investigate which miRNAs were regulated in pulmonary endothelial cells in response to NOD1 and NOD2 stimulation, we used four independent cell isolations with *a priori* confirmed induction of pro-inflammatory genes (Fig 1) and strict criteria for RNA quality control. TaqMan low-density array (TLDA, comprising ~380 unique frequently expressed murine miRNAs, mmu-miRNAs) revealed ~75% not regulated and 10–16% either up- or down-regulated miRNAs after 6 hours of NOD1 and NOD2 stimulation. Total array data are available in the GEO public database (GSE145798) at NCBI. The top 15 differentially expressed microRNAs as ranked by p-value for each condition are presented in a heatmap (Fig 2A) and significantly regulated miRNAs are additionally depicted in a Venn diagram (Fig 2B). All seven identified miRNAs after NOD1 stimulation were found to be consistently down-regulated whereas from the four identified miRNAs after NOD2 stimulation three were down-regulated and one up-regulated. One candidate, miR-147-3p was down-regulated after NOD1 as well as after NOD2 stimulation (Fig 2B).

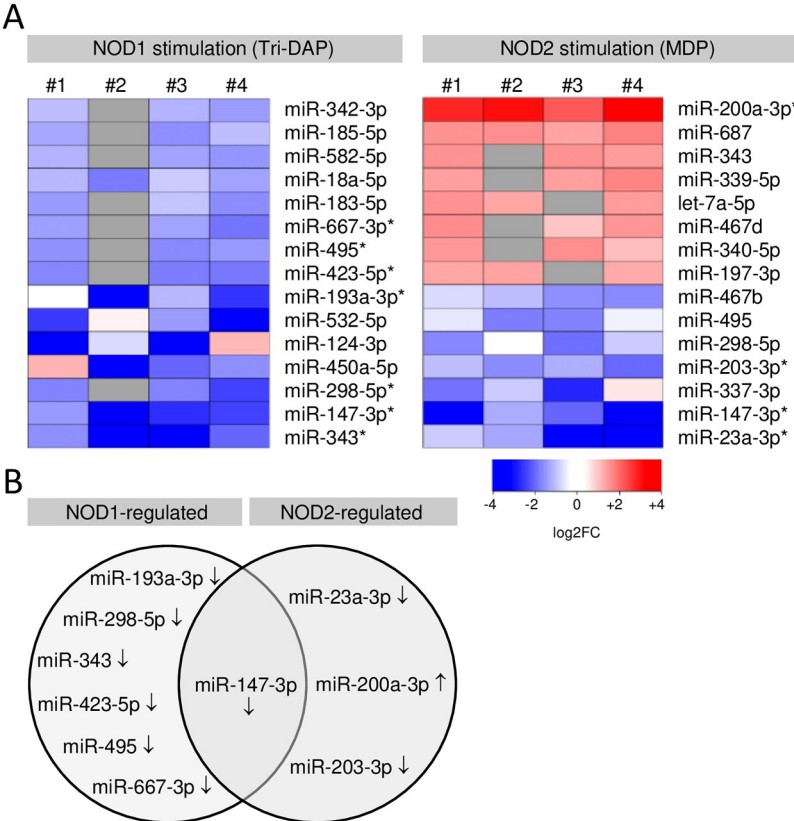

**Fig 2. NOD1- and NOD2-dependently regulated miRNAs in murine pulmonary endothelial cells. (A)** Heatmap showing differentially regulated miRNAs 6 hours following Tri-DAP or MDP stimulation (each 10 μg/mL) from four independent experiments as determined by TaqMan low-density array (TLDA). *P<0.05, grey boxes = below threshold. (**B**) Venn diagram showing significantly differentially regulated miRNAs from the heatmap. ↓ down- ↑ up-regulated vs. unstimulated control.

Subsequently we performed real-time PCR analysis to validate the array data and could confirm significantly reduced expression levels of miR-147-3p and miR-298-5p after NOD1 stimulation and miR-147-3p and miR-200a-3p after NOD2 stimulation in pulmonary endothelial cells (Fig 3A). As a control, we isolated pulmonary endothelial cells from $Nod1^{-/-}$, $Nod2^{-/-}$ double knockout (Nod1/Nod2-KO) mice which consequently did not show differential levels of the identified miRNAs after NOD-dependent stimulation (Fig 3B). In addition, we investigated the miRNA levels after 24 hours of NOD stimulation. Compared to 6 hours, miRNAs were found to be reversely or not regulated (Fig 3C). All candidates identified as down-regulated were significantly enhanced after 24 hours, pointing towards an

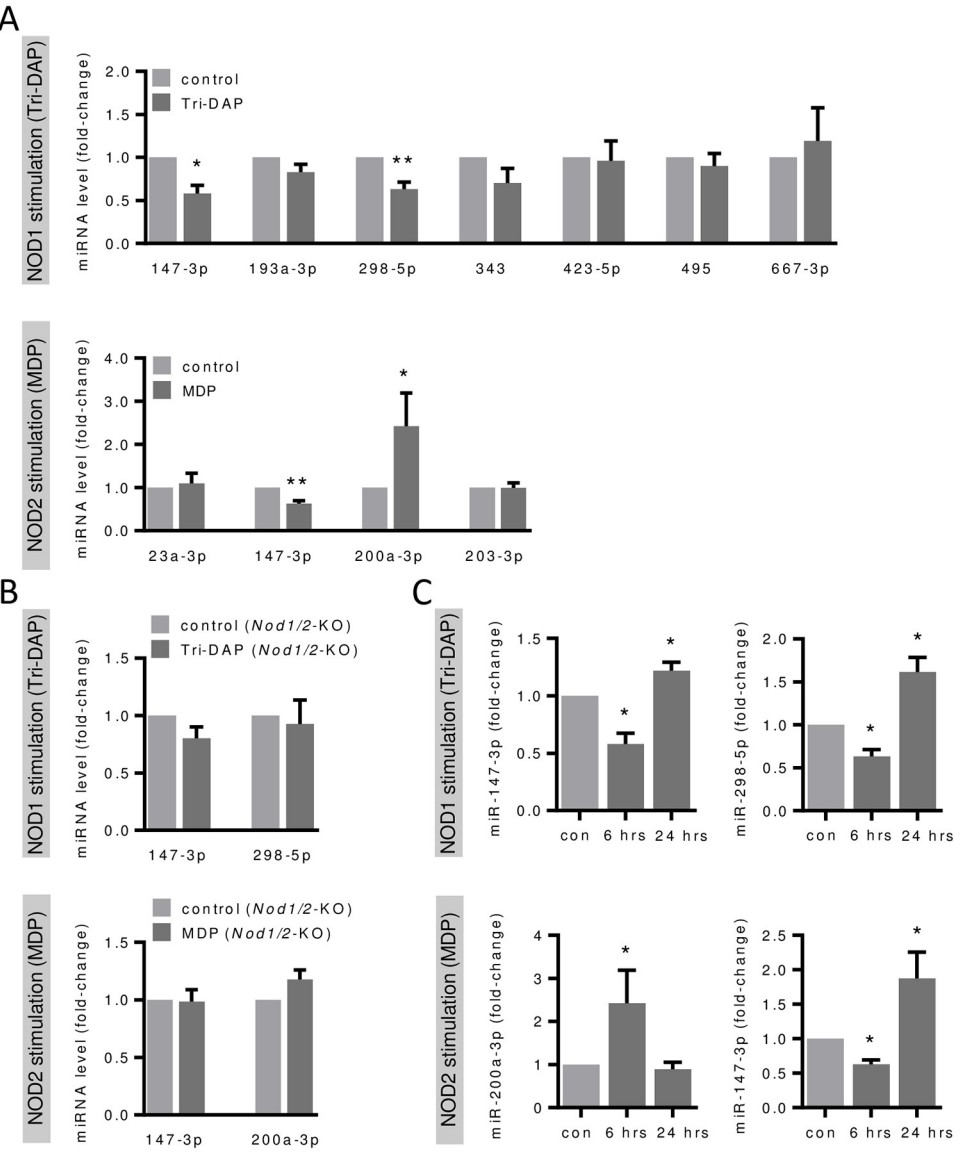

**Fig 3. Validation of NOD1- and NOD2-dependently regulated miRNAs in murine pulmonary endothelial cells.** Expression levels of the indicated miRNAs in cells from (**A**) wild type mice and from (**B**) Nod1/Nod2-KO mice were analyzed 6 hours and from (**C**) wild type mice were analyzed 6 and 24 hours following Tri-DAP or MDP stimulation (each 10 μg/mL) by real-time PCR. *P<0.05, **P<0.01 vs. unstimulated control, n = 3–5.

endogenous negative feedback loop to prevent exuberant and long-lasting translation of their target mRNAs.

## MiR-147-3p and miR-298-5p control NOD1- and NOD2-dependent induction of pro-inflammatory genes in murine pulmonary endothelial cells

Hereafter, we focused on the down-regulated miRNAs miR-147-3p and miR-298-5p, assuming that these candidates are potentially involved in the up-regulation of pro-inflammatory genes in pulmonary endothelial cells following NOD stimulation. TNF-α and IL-6 have been already identified as a target for miR-147 after Toll-like receptor (TLR) stimulation in macrophages [22] and target site analysis predicted binding sites for miR-147-3p and for miR-298-5p (not for 200a-3p) in both 3' UTRs (S2A Fig). Since both candidate miRNAs are rather weakly expressed (CT = ~33) we used an overexpression strategy to explore their potential impact on *Il-6* and *Tnf-α* expression. To endogenously increase the levels of these miRNA candidates, we transfected pulmonary endothelial cells with miRNA mimics, which led to a large enrichment of miR-147-3p and miR-298-5p in comparison to scrambled control transfection (S2B Fig). Transfection of pulmonary endothelial cells with miR-147-3p and miR-298-5p mimics decreased *Tnf-α* mRNA and secreted TNF-α protein levels (Fig 4A and 4B). However, only miR-298-5p mimics were able to decrease *Il-6* mRNA and IL-6 protein levels (Fig 4A and 4B) indicating that miR-147-3p may not be involved in *Il-6* regulation in pulmonary endothelial cells.

To investigate a potential physical interaction of miR-147-3p and miR-298-5p with their target binding sites in HEK-293T cells, we constructed luciferase reporter vectors carrying the 3' UTR of the murine *Tnf-α* and *Il-6* mRNA (Fig 4C). Contrary to mRNA levels in pulmonary endothelial cells (Fig 4A and 4B), transfection of miR-147-3p increased luciferase activity of *Tnf-α* and *Il-6* 3' UTR constructs in HEK-293T cells (Fig 4D). If miR-147-3p is involved in *Tnf-α* and *Il-6* regulation in this context could finally not be elucidated in our study.

In accordance with native mRNA regulation in pulmonary endothelial cells (Fig 4A and 4B), transfection of miR-298-5p decreased luciferase activity of *Tnf-α* and *Il-6* 3' UTR constructs (Fig 4D). Mutation of the miR-298-5p binding site (seed-sequence deletion) restored luciferase activity of *Il-6* but not of *Tnf-α* 3' UTR constructs (Fig 4D) demonstrating that down-regulation of *Il-6* mRNA by miR-298-5p required a direct interaction of the miRNA with the corresponding target binding site in the 3' UTR. To further confirm our results, we additionally used a down-regulation strategy of endogenous miR-298-5p by anti-miRNA. Transfection of pulmonary endothelial cells with anti-miR-298-5p consequently reduced miR-298-5p levels (S2C Fig) and up-regulated *Il-6* mRNA and IL-6 protein levels (Fig 4E and 4F).

## TLR- and NOD-dependent levels of miR-147-3p, miR-200a-5p and miR-298-5p in endothelial cells of different origin

To explore if the miRNA regulation (for all validated candidates, Fig 3A) occurred after stimulation with other PRR agonists as well, we used different TLR agonists to stimulate pulmonary endothelial cells. As expected, TLR2/6, TLR3 and TLR4 stimulation increased *Tnf-α* and *Il-6* mRNA levels, mostly exceeding the effect of NOD stimulation (S3 Fig). However, miRNA regulation after TLR stimulation was clearly different from NOD stimulation, as the levels of all miRNA candidates were found to be enhanced (Fig 5A). Finally, we were interested if the identified miRNA regulation in pulmonary endothelial cells could be also seen in endothelial cells of different origin. In murine cardiac endothelial cells, solely miR-298-5p was found to be down-regulated following NOD1 stimulation, whereas miR-147-3p and 200a-3p levels were

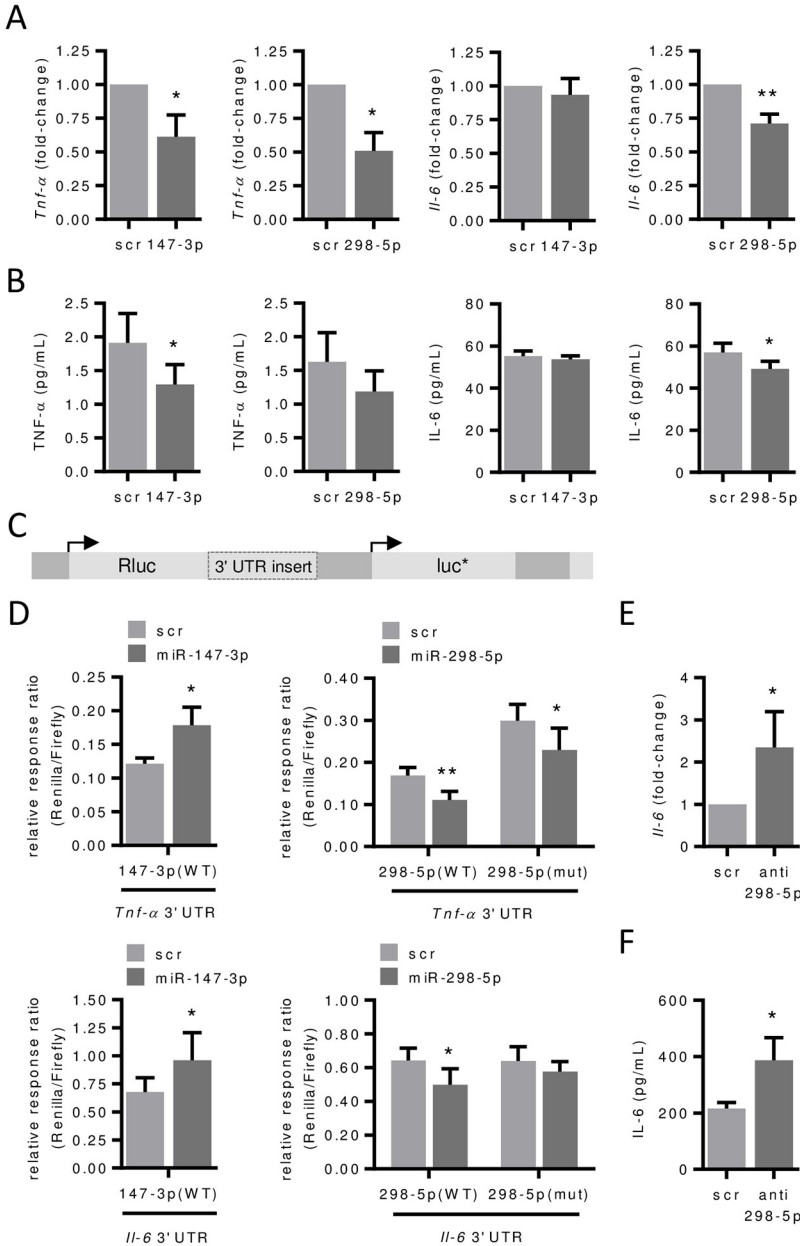

**Fig 4. miR-147-3p and miR-298-5p control NOD1- and NOD2-dependent induction of *Tnf-α* and *Il-6* in murine pulmonary endothelial cells.** (**A**) *Tnf-α* and *Il-6* mRNA levels and (**B**) TNF-α and IL-6 protein levels in the supernatant were analyzed 24 hours after transfection of scramble (scr) control, mimics for miR-147-3p or miR-298-5p (25 pmol each) by real-time PCR or ELISA, respectively. *P<0.05, **P<0.01 vs. scr, n = 4–5. (**C**) Essential elements in the luciferase reporter vector psiCHECK™-2 carrying an insert of the untranslated region (3' UTR) of *Tnf-α* or *Il-6* mRNA sequence downstream of the reporter gene. Rluc = Renilla luciferase, luc* = Firefly luciferase. (**D**) Reporter plasmids with 3' UTR of (wild type = WT), *Tnf-α* or *Il-6* mRNA or its mutated version (= mut, each 200 ng) were cotransfected with scramble (scr) control, mimics for miR-147-3p or miR-298-5p (each 30 pmol) into HEK-293T cells. Renilla luciferase activity was measured after 72 hours and normalized to Firefly luciferase activity. Relative response ratios of Renilla versus Firefly luciferase normalized to empty vector and mimic control are shown *P<0.05, **P<0.01 vs. scr, n = 3–6. (E) *Il-6* mRNA levels and (F) IL-6 protein levels in the supernatant were analyzed 24 hours after transfection of scramble (scr) control or anti-miR-298-5p (25 pmol each) by real-time PCR or ELISA, respectively. *P<0.05, **P<0.01 vs. scr, n = 4–5.

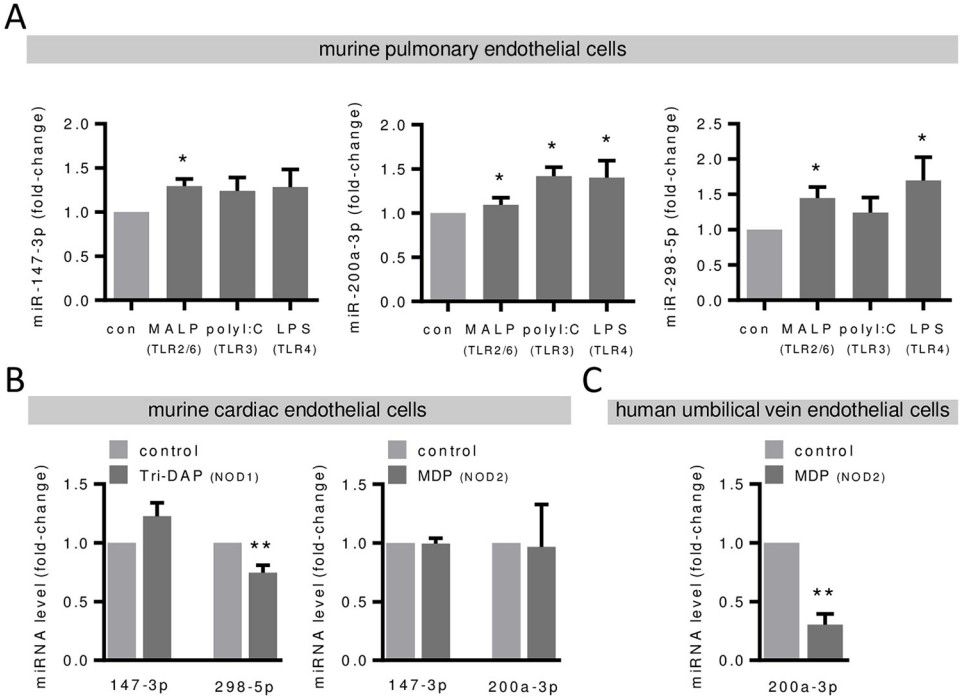

**Fig 5. TLR- and NOD-dependent levels of miR-147-3p, miR-200a-3p and miR-298-5p in different endothelial cells.** MiRNA levels of miR-147-3p, miR-200a-3p and miR-298-5p were analyzed 6 hours after MALP-2 (1 µg/mL), polyI:C (1 µg/mL) or LPS (0.1 µg/mL) stimulation in (**A**) murine pulmonary endothelial cells and after Tri-DAP or MDP stimulation (each 10 µg/mL) in (**B**) murine coronary endothelial cells and in (**C**) human umbilical vein endothelial cells by real-time PCR. $^*P < 0.05$, $^{**}P < 0.01$ vs. unstimulated control, n = 4–6.

not significantly modulated by NOD signaling (Fig 5B). In addition, and contrary to murine pulmonary endothelial cells, we found decreased miR-200a-3p levels following NOD2 stimulation in human umbilical vein endothelial cells (Fig 5C). MiR-147-3p (mmu-miR-147 corresponds to hsa-miR-147b) levels were below the detection limit in these cells and miR-298-5p does not exist in human. These data clearly demonstrate distinct differences in the regulation of the identified miRNAs in murine pulmonary endothelial cells in response to NOD vs. TLR stimulation and in the NOD-dependent regulation in endothelial cells of different origin. However, down-regulation of miR-298-5p by NOD1-stimulation is consistent in murine pulmonary and cardiac endothelial cells.

## Discussion

The function of the endothelium is crucial for vascular homeostasis as well as for the immune response [5,23]. In general, it regulates nutrient and metabolic exchange and the transition of immune cells into the vessel wall and the subjacent tissue. A dysfunctional endothelium does not only occur in bacterial sepsis [24] or *Chlamydia pneumoniae* infection [25] but also critically contributes to the initiation of atherosclerotic vascular disease [25,26]. We have chosen pulmonary endothelial cells for the analysis of NOD-dependently regulated miRNAs, which are in close proximity to infiltrating pathogens from the pulmonary alveolar epithelium and therefore considered as a good model for bacterial infection [5].

Like TLRs [27], NOD1 and NOD2 belong to the PRRs of the immune system for the recognition of a wide variety of bacterial and viral antigens in order to organize the immune defense

via inflammatory cytokines. After receptor ligation, the downstream signalling pathways of NOD1 and NOD2 are highly similar involving a CARD-CARD interaction with the receptor-interacting protein 2 (RIP2) and the transforming growth factor beta-activated kinase 1 (TAK1) and finally leading via activation of NF-κB and MAPK to the induction of inflammatory cytokines such as TNF-α, IL-6, IL-8 and interferon-β (IFN-β) [2]. Transcription factor-dependent induction of cytokine expression following NOD stimulation in various cell types have been well studied, whereas less is known about the role of miRNAs in NOD-dependent induction of cytokine expression. In addition, several miRNAs targeting NOD expression itself in different cell types–including endothelial cells–have been described [11–15]. In regard to cytokine expression, NOD2 signaling up-regulated miR-29 in human dendritic cells and thereby down-regulated IL-23 by targeting IL-12p40 directly and IL-23p19 indirectly [28]. Also in human dendritic cells, NOD2 and IL-12/IL-23p40 were identified as targets of miR-10a [13]. A NOD-dependent induction of pro-inflammatory genes such as IL-1β, IL-6, IL-8 and type I interferons in endothelial cells has been already established [16–18]. However, miR-NAs involved in this process have not been identified so far.

Our screening experiments revealed mainly down-regulated miRNAs following NOD stimulation in pulmonary endothelial cells, presumably, because miRNAs are mainly known as negative regulators of gene expression and reduced miRNA levels normally entail increased mRNA levels. A scenario, that is expected for pro-inflammatory genes during bacterial infection. Accordingly, we observed increased *Tnf-α* and *Il-6* mRNA levels in pulmonary endothelial cells 6 hours after stimulation with NOD ligands. However, the induction was transient and mRNA levels went back to baseline levels after 24 hours. Consequently, miR-147-3p and miR-298-5p –which we identified as potential regulators of NOD-dependent *Tnf-α* and *Il-6* mRNA levels in pulmonary endothelial cells–showed a contrary kinetic. Both miRNAs were down-regulated after 6 hours of NOD stimulation and increased back to baseline levels after 24 hours possibly in order to transiently increase and thereafter limit again the expression levels of *Tnf-α* and *Il-6* mRNA. This process follows a certain logic, since inflammatory responses in general should ideally take place in a temporally limited extend. First, a short-term initial inflammatory response represents a sufficient trigger for the subsequent immune response and second, persisting inflammation may cause severe problems. Especially pathogens in the bloodstream that induce a wave of systemic inflammation could lead to detrimental and possibly fatal sepsis [24,29]. The endothelium has a particularly important role in sensing pathogens and to initiate the immune response but also in keeping exuberant inflammation in check to avoid harm [5]. Moreover, a chronically inflamed and dysfunctional endothelium is a well-known risk factor for the development of atherosclerotic vascular disease [25,26]. In this regard, miR-147 and miR-298 have been already linked to inflammatory gene expression before. MiR-147 has been described as a regulator of TLR-dependent *Tnf-α* and *Il-6* expression in macrophages [22], of aortic inflammation and macrophage activation [30] and of M1 macrophage polarization [31] and miR-298 as a regulator of TNF-α in pancreatic cells [32].

Using luciferase reporter assays with full length 3' UTR reporter constructs of the murine *Il-6* mRNA, we could demonstrate a novel direct interaction of miR-298-5p with the 3' UTR in order to regulate *Il-6* mRNA expression levels. This interaction is lost, if the seed sequence of the identified binding site was deleted. Such a direct interaction was not seen for miR-298-5p within the 3' UTR of the *Tnf-α* mRNA, despite seed sequence deletion of all three predicted binding sites within the 3' UTR. In this regard, our data did not confirm a study from Mor et al. [33], showing a direct interaction of miR-298 with the 3' UTR (large parts but not full-length) of the murine *Tnf-α* mRNA. Site-directed mutagenesis of one single binding site in this study completely prevented binding of miR-298 to the *Tnf-α* 3' UTR. This discrepancy could be explained by detailed differences in plasmid generation and binding site

manipulation. In our study, only a direct interaction of miR-298-5p with the Il-6 3'UTR has been certainly shown by luciferase reporter assays. However, for the other cases, an indirect regulation by yet unidentified intermediate factors is conceivable. Contrary to miR-298-5p, we could not confirm the regulation of *Tnf-α* and *Il-6* via a direct binding by miR-147-3p in luciferase reporter assays. Thus, we could not finally elucidate the role of miR-147-3p in this context.

We are aware that our study has several limitations. First, compared to direct miRNA sequencing the use of TaqMan low-density arrays for selected miRNAs inherently limited the identification of NOD-dependently regulated miRNAs to the candidates present on the array. Next, we were not able to confirm all identified NOD-regulated miRNAs from the array by real-time PCR, a well-known problem in the field. In addition, isolation of primary cells always bears the risk of cross contamination with other cells types, which we try to avoid by performing routinely tests on endothelial markers for cell purity. Moreover, although HEK-293T cells are widely used for luciferase reporter assays, they represent an artificial model, which may explain different regulation of target mRNA levels by candidate miRNAs compared to primary murine pulmonary endothelial cells. Due to inconsistent transfection results, primary cells and different endothelial cell lines were found to be not suitable for investigation of miRNA/target interaction with luciferase reporter assays. We do not want to claim that the observed effects generally apply for all types of endothelial cells. Induced *Tnf-α* and *Il-6* mRNA levels were found to be a general process, independent of endothelial cell origin and PRR agonists (NOD or TLR agonists), whereas we found distinct differences in miR-147-3p, miR-200a-3p and miR-298-5p regulation in different endothelial cell types and with different PRR agonists.

To summarize, following NOD1 and NOD2 stimulation we reported expectedly increased *Tnf-α* and *Il-6* mRNA and protein levels in pulmonary endothelial cells and subsequently identified and confirmed NOD1-dependent regulation of miR-147-3p (↓) and 298-5p (↓) as well as NOD2-dependent regulation of miR-147-3p (↓) and miR-200a-3p (↑) levels. Focusing on the NOD-dependently down-regulated miR-147-3p and miR-298-5p, we explored their function as negative regulators of *Tnf-α* and *Il-6* mRNA levels. We documented that the inhibitory effect of miR-298-5p on *Il-6* mRNA levels is mediated by a direct interaction with the 3' UTR. In conclusion, the present study contributes to promote a better understanding of the NOD-dependent regulation of inflammatory gene expression in pulmonary endothelial cells. NOD-signaling has already been suggested as a novel attractive target in the protection against infections [2] and therefore further studies are needed to investigate how such an approach could be applied in human infectious disease.

## Supporting information

**S1 Fig. Characterization of murine pulmonary endothelial cells and *Nod1* and *Nod2* expression in different endothelial cells.** (**A**) Immunofluorescent staining of murine pulmonary endothelial cells with isolectin B4 and DAPI. A representative picture is shown. Scale bar = 50 μm. (**B**) Cell surface expression of CD31 on murine pulmonary endothelial cells analyzed by flow cytometry. Filled graphs represent CD31 expression, open graphs represent unstained controls. A representative experiment is shown. *Nod1* and *Nod2* mRNA expression was analyzed by real-time PCR in (**C**) murine pulmonary endothelial cells, in (**D**) murine cardiac endothelial cells and NOD1 and NOD2 mRNA in (**E**) human umbilical vein endothelial cells and loaded onto agarose gels. *β-Actin*/β-ACTIN is shown as loading control. Representative pictures are shown. NTC = no template control.
(PDF)

**S2 Fig.** (**A**) Predicted binding sites for miR-147-3p and miR-298-5p in the 3' untranslated region (UTR) of the *Tnf-α* and *Il-6* mRNA sequence. Seed sequences for miR-298-5p, which were deleted for luciferase reporter assays, are underlined and in bold. (**B**) miR-147-3p and miR-298-5p levels 24 hours after transfection of pulmonary endothelial cells with mimics for miR-147-3p or miR-298-5p and scrambled (scr) control (25 pmol each) was analyzed by real-time PCR. **p<0.01 vs. scr, n = 3–4. (**C**) miR-298-5p levels 24 hours after transfection of pulmonary endothelial cells with anti-miR-298-5p and scrambled (scr) control (25 pmol each) was analyzed by real-time PCR, n = 3.
(PDF)

**S3 Fig. Induction of *Tnf-α* and *Il-6* in response to TLR stimulation in murine pulmonary endothelial cells.** Cells were stimulated with the TLR2/6 agonist MALP-2 (1 µg/mL), with the TLR3 agonist polyI:C (1 µg/mL) or with the TLR4 agonist LPS (0.1 µg/mL) and *Tnf-α* and *Il-6* mRNA expression was analyzed after 6 hours by real-time PCR. **P<0.01 vs. unstimulated control (con), n = 5–6.
(PDF)

## Acknowledgments

We thank Daniela Beppler, Silke Brausche, Nadine Siebert and Isabell Beinborn for excellent technical assistance. We thank Siegmund Köhler and his team from University Medical Centre Giessen and Marburg of Philipps University Marburg and all volunteers for providing umbilical cords.

## Author Contributions

**Conceptualization:** Evelyn Vollmeister, Bernhard Schieffer, Karsten Grote.

**Data curation:** Ann-Kathrin Vlacil, Evelyn Vollmeister, Wilhelm Bertrams, Florian Schoesser, Raghav Oberoi, Jutta Schuett, Sonja Huehn, Katrin Bedenbender.

**Formal analysis:** Ann-Kathrin Vlacil, Evelyn Vollmeister, Wilhelm Bertrams, Florian Schoesser, Raghav Oberoi, Jutta Schuett.

**Funding acquisition:** Harald Schuett, Bernd T. Schmeck, Bernhard Schieffer, Karsten Grote.

**Project administration:** Bernhard Schieffer, Karsten Grote.

**Supervision:** Karsten Grote.

**Writing – original draft:** Karsten Grote.

**Writing – review & editing:** Ann-Kathrin Vlacil, Evelyn Vollmeister, Wilhelm Bertrams.

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
