## [Decision Letter · Decision Letter 0]

19 Feb 2020

PONE-D-20-01994

Identification of microRNAs involved in NOD-dependent induction of pro-inflammatory genes in pulmonary endothelial cells

PLOS ONE

Dear Dr. Grote,

Thank you for submitting your manuscript for review to PLoS ONE. I have received to date one completed review (for a total of 6 reviewers contacted) and could not get a second one. I have personally evaluated your article and I totally agree with the comments raised by the reviewer.

After reading the reviews and looking at the manuscript, we feel that your study has merit, but is not suitable for publication as it currently stands. Therefore, my decision is "Major Revision”.

You must revise accordingly and explain your revisions in a covering letter if you wish for us to consider your paper further for publication. Note that it will have to go through another round of review.

We invite you to submit a revised version of the manuscript that addresses the concerns raised by the reviewer. Please pay attention to all the reviewer suggestions and give them due consideration.

Specifically:

You should clearly respond to the three major concerns raised by the Reviewer. In particular, the first major point about miR-147 and/or miR-298 function(s) appears essential and need additional experiments. Moreover, I have also personally raised one additional point: while the authors claim that all data are available, they did not provide the full dataset of miRNA TLDA assay in a specific supplemental table (or a link to a public database).     

We would appreciate receiving your revised manuscript by Apr 04 2020 11:59PM. To enhance the reproducibility of your results, we recommend that if applicable you deposit your laboratory protocols in protocols.io, where a protocol can be assigned its own identifier (DOI) such that it can be cited independently in the future. For instructions see: http://journals.plos.org/plosone/s/submission-guidelines#loc-laboratory-protocols

We look forward to receiving your revised manuscript.

Kind regards,

Bernard Mari, Ph.D

Academic Editor

PLOS ONE

Journal Requirements:

1. Thank you for including your ethics statement; "Human Subject Research

- local ethic regulations of Philipps University Marburg

- AZ 20/16

- written

Animal Research

- all experiments were approved by the governmental animal ethics committee and are conform to the guidelines from directive 2010/63/EU of the European Parliament.

- #50/2015

- for cell isolation, mice at the age of six to ten weeks were sacrificed by cervical dislocation"

With regards to your human ethics statement;

Please amend your current ethics statement to confirm that your named institutional review board or ethics committee specifically approved this study.

Reviewers' comments:

Reviewer's Responses to Questions

**Comments to the Author**

1. Is the manuscript technically sound, and do the data support the conclusions?

Reviewer #1: Partly

2. Has the statistical analysis been performed appropriately and rigorously? 

Reviewer #1: Yes

3. Have the authors made all data underlying the findings in their manuscript fully available?

Reviewer #1: Yes

4. Is the manuscript presented in an intelligible fashion and written in standard English?

Reviewer #1: Yes

5. Review Comments to the Author

Reviewer #1: Please finds enclosed my comments on the manuscript entitled “Identification of microRNAs involved in NOD-dependent induction of pro-inflammatory genes in pulmonary endothelial cells” by Vlacil et al.

This article describes a link between NOD1/2-regulated microRNAs (miRs) and NOD1/NOD2-dependent inflamatory response (IL6/TNF) in pulmonary endothelial cells. The authors found that concomitantly to IL6 and TNF regulation by NODs stimulation miR-147 and miR-298 are transiently downregulated by NODs stimulation. Therefore, using gain of function assay they suggest that these miRs can be involved in the transient induction of IL6 and TNF by NODs stimulation.

Finaly the authors try to emphasize the specificity of their mechanisms (NOD-dependent mechanism)

The manuscript is well written and is interesting in some regards as it adress for the first time the miRNA regulation by the NOD pathway in pulmonary endothelial cells. The first part of the manuscript (identification of NOD-dependent miRNA regulation) is well designed and executed with several controls, including NODs KO cells. However, the second part of the manuscript that aim to decipher the mechanism by which miR-147 and miR-298 participate to the NOD signaling pathway remains poorly address

Therefore several concerns need to be address before is publication.

Major:

1) The authors state that the expression level of miR-147 and miR-298 is low (CT=33), therefore, they only use overexpression experiments to demonstrate the role of these miRNA in the regulation of IL6 and TNF. However the authors also claim that the downregulation of miR-147 and miR-298 participate to the upregulation of IL6 and TNF. Therefore the authors should perform loss of function experiments to demonstrate that the downregulation of miR-147 and/or miR-298 participate to the control of TNF/IL6 expression (mRNA and protein). Alternatively the authors could also try to perform experiment in which they overexpressed miR-298 in conbination with target blocking site(s) oligos

2) The authors should cleary state that the mechanism by which miR-147 participates to the control of IL6/TNF remains enigmatic

3) The authors should remove "In silico analysis" from the abstarct as no in silico analysis were performed. At least for the reviewer, looking at putative binding sites in the 3'UTR of two transcripts do not account for in silico analysis.

Minor:

1) In Figure 2 downregulated miRNA are in red while up-regulated miRNAs are in blue. It is confusing as in general up-regulated genes/miRNA are represented in red while downregulated genes are represented in blue.

2) In the text the authors should cleary state that miR-147 is mmu-miR-147 which correspond to hsa-miR-147b. It is important, as the existence of hsa-miR-147a (miR-147) remains debate

3) The authors should acknoledge previous work on mmu-miR-147/hsa-miR-147b and the inflamatory responses.

6. PLOS authors have the option to publish the peer review history of their article (what does this mean?). If published, this will include your full peer review and any attached files.

Reviewer #1: No

---

## [Author Response · Author response to Decision Letter 0]

17 Mar 2020

Response to Reviewers

We highly appreciate the valued suggestions and criticism of the reviewer and the editor in order to strengthen the impact of our manuscript. Please find below a detailed response to all comments. Please note that we have added some replicates to single experiments in the course of the revision which leads to minor changes in bar graphs and significant values.

Reviewer #1:

Reviewer: The authors state that the expression level of miR-147 and miR-298 is low (CT=33), therefore, they only use overexpression experiments to demonstrate the role of these miRNA in the regulation of IL6 and TNF. However, the authors also claim that the downregulation of miR-147 and miR-298 participate to the upregulation of IL6 and TNF. Therefore, the authors should perform loss of function experiments to demonstrate that the downregulation of miR-147 and/or miR-298 participate to the control of TNF/IL6 expression (mRNA and protein). Alternatively, the authors could also try to perform experiment in which they overexpressed miR-298 in combination with target blocking site(s) oligos.

Authors: We used different methods, i.e. TaqMan low-density array, transfection of mimics and luciferase reporter assays to identify NOD-dependently expressed miRNAs to investigate for Tnf-α and Il-6 mRNA regulation in murine pulmonary endothelial cells. Finally, we could confirm miR-298-5p from a couple of initially identified miRNAs by all methods and a direct physical interaction in case of Il-6. We have repeatedly stated this issue in the manuscript.

Following the remarks of the reviewer, we consequently focused on miR-298-5p and performed the suggested loss of function experiments using antagomirs for miR-298-5p and were able to further confirm our findings (result section on page 16/17 and novel Figure 4E and 4F and supplemental Figure S2C).

Reviewer: The authors should clearly state that the mechanism by which miR-147 participates to the control of IL6/TNF remains enigmatic

Authors: We explicitly stated now that we could not finally elucidate if miR-147-3p is involved in Tnf-α and Il-6 regulation in pulmonary endothelial cells in this context (result section on page 16 and discussion section on page 20).

Reviewer: The authors should remove "In silico analysis" from the abstract as no in silico analysis were performed. At least for the reviewer, looking at putative binding sites in the 3'UTR of two transcripts do not account for in silico analysis.

Authors: We have deleted the term "In silico analysis" from the abstract, method and result sections.

Reviewer: In Figure 2 downregulated miRNA are in red while up-regulated miRNAs are in blue. It is confusing as in general up-regulated genes/miRNA are represented in red while downregulated genes are represented in blue.

Authors: We have changed the colors in the heatmap in Figure 2A according to the reviewers´ suggestion.

Reviewer: In the text the authors should clearly state that miR-147 is mmu-miR-147 which correspond to hsa-miR-147b. It is important, as the existence of hsa-miR-147a (miR-147) remains debate.

Authors: We have added this information to the result section on page 17.

Reviewer: The authors should acknowledge previous work on mmu-miR-147/hsa-miR-147b and the inflammatory responses.

Authors: Since we investigated both NOD-dependently down-regulated candidates from the initial array – miR-147 and miR-298 – we added additional previous work on inflammatory responses for both miRNAs (ref# 30-32) to the discussion section on page 20.

Editor:

Editor: You should clearly respond to the three major concerns raised by the Reviewer. In particular, the first major point about miR-147 and/or miR-298 function(s) appears essential and need additional experiments. 

Authors: We addressed all concerns raised by the reviewer including additional loss of function experiments on miRNA function, please see above.

Editor: Moreover, I have also personally raised one additional point: while the authors claim that all data are available, they did not provide the full dataset of miRNA TLDA assay in a specific supplemental table (or a link to a public database). 

Authors: We now provided the full dataset in the GEO public database (GSE145798) at NCBI and included this information in the method section on page 10 and in the result section on page 13.

At the moment, results are deposited as ‘private data’. In case of acceptance, we will set the status to ‘public data’. The reviewer may use: https://www.ncbi.nlm.nih.gov/geo/query/acc.cgi?acc=GSE145798 and token: atofyckibhmnnwt to view the full dataset.

Journal Requirements:

Please amend your current ethics statement to confirm that your named institutional review board or ethics committee specifically approved this study.

Authors: We have amended our ethics statement on page 6 in the method section as requested.

---

## [Decision Letter · Decision Letter 1]

11 Apr 2020

PONE-D-20-01994R1

Identification of microRNAs involved in NOD-dependent induction of pro-inflammatory genes in pulmonary endothelial cells

PLOS ONE

Dear Dr. Grote,

Thank you for resubmitting your manuscript for review to PLoS ONE. While you have adequately addressed the queries in the review and that the revised manuscript is significantly improved from its original submission, we feel that your manuscript still needs a few improvements. Therefore, my decision is "Minor Revision.

As requested by the reviewer, the authors should replace the term antagomiR by anti-miRNA and should include the manufacturer of their anti-miRNA. 

We would appreciate receiving your revised manuscript by May 26 2020 11:59PM. To enhance the reproducibility of your results, we recommend that if applicable you deposit your laboratory protocols in protocols.io, where a protocol can be assigned its own identifier (DOI) such that it can be cited independently in the future. For instructions see: http://journals.plos.org/plosone/s/submission-guidelines#loc-laboratory-protocols

We look forward to receiving your revised manuscript.

Kind regards,

Bernard Mari, Ph.D

Academic Editor

PLOS ONE

Reviewers' comments:

Reviewer's Responses to Questions

**Comments to the Author**

1. If the authors have adequately addressed your comments raised in a previous round of review and you feel that this manuscript is now acceptable for publication, you may indicate that here to bypass the “Comments to the Author” section, enter your conflict of interest statement in the “Confidential to Editor” section, and submit your "Accept" recommendation.

Reviewer #1: All comments have been addressed

2. Is the manuscript technically sound, and do the data support the conclusions?

Reviewer #1: Yes

3. Has the statistical analysis been performed appropriately and rigorously? 

Reviewer #1: Yes

4. Have the authors made all data underlying the findings in their manuscript fully available?

Reviewer #1: Yes

5. Is the manuscript presented in an intelligible fashion and written in standard English?

Reviewer #1: Yes

6. Review Comments to the Author

Reviewer #1: The authors did a good job.

I only have a minor comment.

The authors should replace the term antagomiR by anti-miRNA, as antagomiR refer to a specific class of miR inhibitors (coupled to cholesterol allowing miRNA delivery without adding transfection readgent)

In addition, the authors should include the manufacturer of their anti-miRNA

7. PLOS authors have the option to publish the peer review history of their article (what does this mean?). If published, this will include your full peer review and any attached files.

Reviewer #1: Yes: Thomas BERTERO

---

## [Author Response · Author response to Decision Letter 1]

14 Apr 2020

Response to Reviewers

Reviewer #1:

Reviewer: The authors should replace the term antagomiR by anti-miRNA, as antagomiR refer to a specific class of miR inhibitors (coupled to cholesterol allowing miRNA delivery without adding transfection reagent).

In addition, the authors should include the manufacturer of their anti-miRNA.

Authors: We have replaced the term antagomir by anti-miRNA throughout the manuscript and included the manufacturer (Dharmacon as for mimics) on page 9 of the material and method section.

---

## [Editor Report · Decision Letter 2]

17 Apr 2020

Identification of microRNAs involved in NOD-dependent induction of pro-inflammatory genes in pulmonary endothelial cells

PONE-D-20-01994R2

Dear Dr. Grote,

We are pleased to inform you that your manuscript has been judged scientifically suitable for publication and will be formally accepted for publication once it complies with all outstanding technical requirements.

With kind regards,

Bernard Mari, Ph.D

Academic Editor

PLOS ONE
---

## [Editor Report · Acceptance letter]

21 Apr 2020

PONE-D-20-01994R2 

Identification of microRNAs involved in NOD-dependent induction of pro-inflammatory genes in pulmonary endothelial cells 

Dear Dr. Grote:

I am pleased to inform you that your manuscript has been deemed suitable for publication in PLOS ONE. Congratulations! Your manuscript is now with our production department. 

With kind regards,

on behalf of

Dr. Bernard Mari 

Academic Editor

PLOS ONE